# Self-Regulating Adaptive Controller for Oxygen Support to Severe Respiratory Distress Patients and Human Respiratory System Modeling

**DOI:** 10.3390/diagnostics13050967

**Published:** 2023-03-03

**Authors:** Indrajit Naskar, Arabinda Kumar Pal, Nandan Kumar Jana

**Affiliations:** Heritage Institute of Technology, Kolkata 700107, WB, India

**Keywords:** respiratory distress patient, respiratory failure, model reference adaptive control, set-point modulated fuzzy-based control, mathematical modeling of the human respiratory system with exchange of oxygen

## Abstract

Uncontrolled breathing is the most critical and challenging situation for a healthcare person to patients. It may be due to simple cough/cold/critical disease to severe respiratory infection of the patients and resulting directly impacts the lungs and damages the alveoli which leads to shortness of breath and also impairs the oxygen exchange. The prolonged respiratory failure in such patients may cause death. In this condition, supportive care of the patients by medicine and a controlled oxygen supply is only the emergency treatment. In this paper, as a part of emergency support, the intelligent set-point modulated fuzzy PI-based model reference adaptive controller (SFPIMRAC) is delineated to control the oxygen supply to uncomforted breathing or respiratory infected patients. The effectiveness of the model reference adaptive controller (MRAC) is enhanced by assimilating the worthiness of fuzzy-based tuning and set-point modulation strategies. Since then, different conventional and intelligent controllers have attempted to regulate the supply of oxygen to respiratory distress patients. To overcome the limitations of previous techniques, researchers created the set-point modulated fuzzy PI-based model reference adaptive controller, which can react instantly to changes in oxygen demand in patients. Nonlinear mathematical formulations of the respiratory system and the exchange of oxygen with time delay are modeled and simulated for study. The efficacy of the proposed SFPIMRAC is tested, with transport delay and set-point variations in the devised respiratory model.

## 1. Introduction

At present, respiratory distress may be due to different respiratory tract-infected viruses, which are mainly rhinoviruses and enteroviruses (Picornaviridae), influenza viruses (Orthomyxoviridae), parainfluenza, metapneumoviruses and respiratory syncytial viruses (Paramyxoviridae), coronaviruses (Coronaviridae), several adenoviruses, or maybe some critical medical conditions. Out of the above viruses, adenoviruses have a DNA genome, and all others possess an RNA genome [1]. They are usually transmitted by direct hand-to-surface-to-hand contact or aerosol inhalation and replicate in both the upper and lower airways. 

Breathing exercise issues due to any one of the above are one of the most uncontrollable and uncomfortable conditions of humans due to any cause, such as chronic obstructive pulmonary disease(COPD), bronchitis, emphysema, fibrosis, asthma, any medical critical care condition, and so on. There also are other physical conditions with acute respiratory infection symptoms in particular for older people and immune-suppressed patients such as fatigue, reduced alertness, reduced mobility, and many more [2,3]. As a result, there is a probability to reduce the oxygen saturation for the different above conditions of the patients. A longer duration of this condition may also extend to the end of an individual’s life. In the damaged area of the respiratory tract, there is a secretion that is a plasma protein that accumulates on the alveolus wall and thickens the lining. This produces a blockage in the path of oxygen transfer to the red blood cells, resulting in difficulty breathing, which creates a deficiency of oxygen in the internal organs. In this condition, the deficiency of oxygen in the human body, malfunctioning of organs, and also immunity deficiency enhance the present condition, which aggravates the crisis [4,5]. Mehedi, Ibrahim et al. [6] implemented a fuzzy PID controller for mechanical ventilation processes. Tuning a fuzzy controller is a strenuous task due to the substantial number of control parameters involved. But Mehedi, Ibrahim et al. [6] did not tune the controller, which is very much essential for critical processes such as the human respiratory system. Hansen et al. [7] proposed the O2matic^®^-based closed-loop controlled oxygen supply in COVID time, which has limitations of flow rate and it adjusts flow every second based on 15 s averaging of SpO_2_ reading.

To sort out this emergency crisis, a scheme is proposed in this paper to design a modified model-based adaptive controller that can automatically control the oxygen level with a faster response but without overshoot, which is also the most important part of the designed controller to protect the patient from the over-concentration of oxygen. To suppress the above problem, a set-point modulated fuzzy PI-based model reference adaptive controller (SFPIMRAC) is proposed to control the oxygen level of respiratory distress patients without human intervention. The block diagram of the proposed scheme is shown in Figure 1. 

## 2. Proposed Components of Respiratory Distress Treatments

The basic components of the proposed model to control the oxygen supply to critical respiratory distress patients are briefed in Figure 1. In this critical time, patients with respiratory distress need to be treated with oxygen therapy to restore normal breathing and also receive proper medication support. The proposed model could be an answer to this problem due to its automatic control capability of oxygen supply to patients without much human intervention. Mathematical modeling of the proposed system components, as listed in Figure 1, is derived below for the simulation study.

### 2.1. Oxygen Cylinder/Oxygen Concentrator

Oxygen therapy-related data with signs of severe respiratory distress, hypoxemia (i.e., SpO_2_ < 90%), or shock are initiated oxygen therapy at 5 L/min and titrated to SpO_2_ ≥ 90% in non-pregnant adults, and it will be SpO_2_ ≥ 92–95% in pregnant patients. A pulse oximeter measures the SpO_2_ level and checks the availability of an oxygen delivery system in the patient care unit [3].

Ideally, in any hospital, oxygen is supplied to the patients from the dedicated oxygen plant only for medical applications with a percentage of O_2_ (not less than 99.5% *v*/*v* of O_2_) concentration in that hospital through piping as shown in Figure 2, but unfortunately, most hospitals and nursing homes do not have this facility, and thus they purchase oxygen cylinders from private vendors for treatment. 

In this paper, a typical oxygen cylinder is modeled for the simulation study. In Figure 3, the variables P_1_, P_2_, q, R, and C are the input oxygen pressure (from the oxygen cylinder), output oxygen pressure (to the patient), oxygen flow rate, resistance in the flow path (valve resistance), and capacity (patient oxygen requirement) of the system, respectively. Considering the above parameters, the mathematical equation is formed using the electrical analogy.
(1)$$\frac{P_{1}-P_{2}}{R}=C\frac{{\mathrm{dP}}_{2}}{\mathrm{dt}}$$

The transfer function of the oxygen cylinder (TFcy) is derived by the Laplace transform of Equation (1).
(2)$$TF_{cy}=\frac{P_{2}\left(S\right)}{P_{1}\left(S\right)}=\frac{1}{RCS+1}=\frac{1}{\tau \;S+1}$$

Equation (2) states that if a patient has severe breathing issues and needs more oxygen, then the C-value will be higher, which will make the R-value lower (i.e., the percentage of valve opening connected to the oxygen cylinder will be higher) or vice versa. In the present work, the value of the time constant (τ = CR) for the simulation study is considered as 0.50 s. and its reason is studied in the result section; however, it can work for other values also. The regulation of oxygen supply is manipulated by the valve opening (here, a change in R-value).

An instant supply of oxygen is very much essential for the treatment of uncontrolled acute breathing problems and thus needs immediate attention to improve and stabilize the SpO_2_ level. In emergency times, when traditional compressed-oxygen cylinders provide a limited amount of oxygen, it is better to equip each hospital with an oxygen concentrator/oxygen plant. Oxygen concentrators operate by drawing air from the environment to deliver a clean, continuous flow at a concentration of oxygen in the range of 82–96%. It would be better to use a high-efficiency particulate air (HEPA) filter with an oxygen concentrator to create a more protected environment for the patients. The Institute of Environmental Sciences and Technology dictates that a HEPA filter must trap 99.97% of particulates of 0.01 microns or larger, whereas the average size of any virus is not less than around 0.123 microns [3].

### 2.2. Pulse Oximeter

A pulse oximeter is used to measure the oxygen concentration in a patient’s blood in a non-invasive manner. The output of the pulse oximeter signal is fed to the comparator to monitor oxygen therapy in this proposed model, as outlined in Figure 1 [3].

## 3. Mathematical Model of Respiratory System

The airway of the human respiratory system, comprising the nose, mouth, pharynx, larynx, trachea, bronchi, and alveoli, carries air between the lungs and the body’s exterior. In this paper, the human respiratory system is modeled in two halves. In the first part of the model, the flow of oxygen from the oxygen concentrator/cylinder to the alveoli passing through the nasal cavity, trachea, and bronchi is considered [6,7,8]. Gas exchange in the human respiratory system is modeled in the next part [9,10,11]. Moreover, a variable transport delay is included in the model to increase effectiveness.

The critical respiratory infection makes the alveolar sacs in the lungs stiff and thick. This degraded condition of the tissues makes it harder for O_2_–CO_2_ exchange to take place through the walls of the alveoli with the bloodstream. Thus, the alveolar sacs stiffen, and as a result, the characteristic compliance of the alveoli decreases [10].

### 3.1. Respiratory Model Part I

The human respiratory system is broadly divided into four subgroups, starting with the nasal cavity, as shown in Figure 4. The next three subgroups, i.e., trachea-bronchial trees, are further subdivided into 24 generations (Table 1) [12,13,14]. In Table 1, generation ‘0’, generation 1 to 19, and generation 20 to 23 correspond to the trachea, bronchi, and alveolar sacs, respectively. In this model, each branch of a given generation is further subdivided into two identical daughters; therefore, generation ‘n’ has 2^n^ branches [14,15]. The electrical equivalence of each generation in terms of resistance (R), inertance (L), and compliance (C) is derived as shown in Table 1 [13,16].

The transfer functions of the respiratory parts are derived from their electrical equivalent model, as depicted in Figure 4. The corresponding values in terms of R, L, and C, and their combinations, are presented in Table 2 [17,18,19]. Different sub-groups of Figure 4 are then modeled in terms of transfer functions as follows: (3)$${\mathrm{TF}}_{N}=1/\left(2.7\times 10^{-3}S^{2}+2.165S+1\right)$$
(4)$${\mathrm{TF}}_{T}=1/\left(3.7\times 10^{-4}S^{2}+5.4\times 10^{-3}S+1\right)$$
(5)$${\mathrm{TF}}_{B}=1/\left(1.44\times 10^{-5}S^{2}+4.02\times 10^{-4}S+1\right)$$
(6)$${\mathrm{TF}}_{A}=1/\left(6.72\times 10^{-8}S^{2}+5.71\times 10^{-4}S+1\right)$$

#### Change in Respiratory Model Due to Defect in Alveoli Area

Abnormality in the alveoli area can be reflected in the developed model in terms of a decrement in the capacitance value of the alveoli by about 100 to 1000 times, which means that the effective impedance $Z=\sqrt{R^{2}+{\left(X_{L}-X_{C}\right)}^{2}}$ of the circuit will increase. The model of the alveolar section represented by Equation (6) thus gets modified to:(7)$${\mathrm{TF}}_{A}^{\prime }=1/\left(1.0\times 10^{-10}S^{2}+1.15\times 10^{-6}S+1\right)$$ where ${\mathrm{TF}}_{A}^{\prime }$ refers to the transfer function of the alveolar section of a respiratory distress patient being affected by any of the above-mentioned viruses/critical medical conditions.

The resultant model, TF_M_ is derived from the series of combinations of the individual models, i.e.,
(8)$${\mathrm{TF}}_{M}={\mathrm{TF}}_{N}*{\mathrm{TF}}_{T}*{\mathrm{TF}}_{B}*{\mathrm{TF}}_{A}^{\prime }$$

### 3.2. Respiratory Model Part II (Gas Exchange)

From Figure 5, it is seen that the gas exchange takes place in the area of alveolar air space, lung tissue, and capillary blood accordingly. In the case of the diffusion process, the movement of the gas from a high-concentration region to a low-concentration region is followed by the gradient profile [19]. This principle is followed during respiration, where gas exchange occurs between the blood and lungs. During this time, the concentration of oxygen is high and the carbon dioxide concentration is low in the blood, which undergoes the gas exchange with air in the lungs with a higher gradient profile than the blood. This process is the opposite of the expiration process [10,11,20]. Each of the alveoli is surrounded by a network-like structure of very small blood vessels of diameter 5 to 10 μm. For a normal person, under resting conditions, the volume of air available for gas exchange corresponds to the alveolar zone, which sums up to around 2.5 to 3 L, and the volume of blood in the neighborhood of the exchange surface is around 70 mL [21,22].

### 3.3. Partial Pressure Model

The concept of partial pressure is adopted to develop the mathematical model of the gas exchange part of the respiratory system. Mathematically, to find out the individual components’ concentration in a mixture of gases, the partial pressure measuring method is used here, and the total pressure is calculated by the sum of the individual components [10,19]. The rate of diffusion of a gas is measured by its partial pressure within the total gas mixture, and they are linearly related. In this gas exchange model, the partial pressures for the alveolar air, lung tissue, and capillary blood are illustrated by $po_{2A}$,$\;po_{2T}$, and $po_{2B}$ respectively [20]. The diffusion of oxygen, as shown in Figure 5, across lung tissue/alveolar air barriers and the lung tissue/capillary blood is modeled as follows:(i)Oxygen diffusion from alveolar air space to lung tissue:
(9)$$\frac{dpo_{2A}}{dt}=\frac{D_{TA}}{\sigma_{A}V_{A}}\left(p_{o_{2T}}-p_{o_{2A}}\right)$$
(ii)Similarly, oxygen exchange between lung tissue, alveolar air space, and capillary blood is derived by:
(10)$$\frac{dpo_{2T}}{dt}=\frac{D_{TB}}{\sigma_{T}V_{T}}\left(p_{o_{2B}}-p_{o_{2T}}\right)+\frac{D_{TA}}{\sigma_{T}V_{T}}\left(p_{o_{2A}}-p_{o_{2T}}\right)$$
(iii)Oxygen transfer between lung tissue and capillary blood:
(11)$$\frac{dpo_{2B}}{dt}=\frac{D_{TB}}{\sigma_{B}V_{B}}\left(p_{o_{2T}}-p_{o_{2B}}\right)$$
*V_A_*, *V_T_*, and *V_B_* are the respective volumes of alveolar air space, lung tissue, and capillary blood. The parameters $\sigma_{A},\;\sigma_{T},$ and $\sigma_{B}$ are used to convert the partial pressure of oxygen in alveolar air space, lung tissue, and capillary blood regions to their corresponding molar concentrations, respectively. In the model, the diffusion rates for lung tissue to alveolar air and lung tissue to capillary blood are illustrated by $D_{TA}$ and $D_{TB}$, respectively. The different parameter values of the model are presented in Table 3. The gas exchange with the capillary blood is a very complex process. Ideally, different biological processes are functioning in this process of gas exchange, such as diffusion of oxygen and carbon dioxide, hemoglobin uptake of oxygen, and enzymatic reactions governing carbon dioxide and bicarbonate levels [10,20]. However, to make the model simple, in this work, only the rate of oxygen transfer is assumed, and some assumptions are considered as follows:

Assumptions: (i) Diffusion takes place in only two zones: capillary blood/lung tissue and lung tissue/alveolar air space. (ii) The alveolar air space must be well mixed. (iii) The same diffusion rate is considered here for each section of the gas exchange module (iv) and the continuous supply of oxygen is achieved by the ventilation process that provides fresh oxygen to the lungs, whereas venous blood is periodically pumped onto the exchange zone by the heart.

#### Electrical Analogy Model

An electrical equivalent circuit of Figure 5 is drawn in Figure 6.

Considering the flow of oxygen as current flow, the following equations are derived from Figure 6:(i)Flow of oxygen from the alveolar air space (AA) to the lung tissue (LT):
(12)$$\frac{dv_{AA}}{dt}=\frac{v_{LT}-v_{AA}}{R_{A}C_{A}}$$

(ii)Oxygen exchange between lung tissue (LT), alveolar air space (AA), and capillary blood (CB):


(13)
$$\frac{dv_{LT}}{dt}=\frac{v_{CB}-v_{LT}}{R_{T}C_{T}}+\frac{v_{AA}-v_{LT}}{R_{A}C_{T}}$$


(iii)Oxygen transfer between LT and CB:


(14)
$$\frac{dv_{CB}}{dt}=\frac{v_{LT}-v_{CB}}{R_{B}C_{B}}$$


The following equation parameters are derived from a careful comparison of Equations (12) to (14) with (9) to (11) and the data in Table 3:R_A_C_A_ = 1/0.2429, R_A_C_T_ = 1/4.76,
R_T_C_T_ = 1/15.87, and R_B_C_B_ = 1/88.88

The ultimate gas exchange model (TF_BO2_) due to the diffusion of oxygen in the capillary blood is calculated by the Laplace transform of Equations (12) to (14). The transfer function (TF_BO2_) is derived by putting the corresponding resistance and capacitance values in Equation (15).
(15)$${\mathrm{TF}}_{{\mathrm{BO}}_{2}}=\left(20s^{2}+200s+62\right)/\left(s^{3}+110s^{2}+350s+67\right)$$

The ultimate human respiratory system (TP_M_) in the infected area of the alveoli of the patient is modeled by a series combination of the models (TF_M_) and (TF_BO2_).
(16)$$TP_{M}=TF_{M}\times TF_{BO_{2}}$$

The stability of the model is checked by the Bode plot in Figure 7. The positive values of gain margin (G_m_) and phase margin (P_m_) ensure that the respiratory model is a stable system. 

### 3.4. Transport Delay

To make the model more realistic, in this study, a variable transport delay is included between the respiratory model and the gas exchange model. The study is carried out with various transport delays, as the time taken in gas exchange is not fixed in respiratory infected patients; it varies depending on the severity. 

## 4. Design of the Proposed Controller

In this section, the design part of the proposed set-point modulated fuzzy PI model reference adaptive controller (SFPIMRAC) is elaborated. Initially, the developed respiratory model for the respiratory distress patient is tested with a conventional PID controller and MRAC to control the oxygen supply and later MRAC is modified by incorporating the knowledge of fuzzy logic with a set-point tracking facility.

### 4.1. Design of MRAC

The basic concept of the MIT rule is applied to the design of the MRAC to control the oxygen supply to respiratory distress patients with respiratory infections associated with breathing problems. A basic scheme of MRAC is illustrated in Figure 8. 

Optimization of control parameters to minimize the loss function is very important in any controller development. In this paper, the difference between patients’ output (*y*) (pulse oximeter reading) and the model process output (*y_m_*) is measured in terms of error (*e*) as pictured in Figure 8. 

In the case of MRAC, there are two parameters: the adaptation gain (*γ*_1_) control and adjustable control parameter (*θ*) is the most important part of the design. A suitable adjustment mechanism of (*θ*) that makes the measured error (*e*) to a value zero is also used to minimize the loss function *F*(*θ*), as shown in Figure 8 [23,24,25].
(17)$$F\left(\theta \right)=\frac{e^{2}}{2}$$

Minimization of loss function *F*(*θ*) is derived from the variation of the parameter θ in the direction of the negative gradient of *F*(*θ*) and adaptation gain (*γ*_1_), i.e.,
(18)$$\frac{d\theta }{dt}=-\gamma_{1}\frac{\partial F}{\partial \theta }=-\gamma_{1}e\frac{\partial e}{\partial \theta }$$
where *∂e/∂θ* is expressed as the sensitivity of the system. For the large process, there are many numbers of process variables such as (*θ*_1_, *θ*_2_, etc.) where Equation (18) can also be applicable. But in that case, output (*θ*) should be taken as a vector quantity in place of scalar one and mathematically the gradient of the error concerning the parameters concept is used to calculate its partial derivative *∂e/∂θ* [24,26]. In Figure 8, the patient and the reference models are represented by *KG*(*s*) and *K*_0_*G*(*s*), respectively (where *K* is an unknown and *K*_0_ is a known parameter). The output of the patient (*y*) always tracks the reference model output (*y_m_*). From the scheme shown in Figure 8, the process output (*y*), model output (*y_m_*), and error (*e*) are derived as follows:(19)y=u.KG(s)=r.θ.KG(s)
(20)$$y_{m}=rK_{0}G\left(s\right)$$
(21)$$e=y-y_{m}=r\theta KG\left(s\right)-rK_{0}G\left(s\right)$$

From (20):$$rG(S)=\frac{y_{m}}{K_{0}}$$
where *u* is the controller output and *r* is the input to the reference model.

The calculation of *∂e/∂θ* is derived by taking a partial derivative of (21).
(22)$$\frac{\partial e}{\partial \theta }=rKG\left(s\right)=K.\frac{y_{m}}{K_{0}}=\left(K/K_{0}\right)y_{m}$$

Finally, the equation for adjusting the parameter variation in (18) is modified by (22).
(23)$$\frac{d\theta }{dt}=-\gamma_{1}e\frac{\partial e}{\partial \theta }=\left(\gamma_{1}K/K_{0}\right)y_{m}e=-\gamma y_{m}e$$

From Figure 8, it is observed that the final output of the controller, or basically input to the process u [*u = rθ*] is obtained by the product of the reference input (*r*) with the adjustable control parameter (*θ*), which is obtained by integrating (*dθ/dt*). In Equation (23), the value of adaptation gain (*γ*_1_) is a user-defined parameter that is a positive number, and its value depends on the process model [27,28,29,30]. 

The choice of exact adaptation gain (*γ*_1_) is one of the most difficult steps in MRAC. To eliminate this difficulty, a set-point modulated fuzzy PI-based scheme is incorporated with MRAC to control the oxygen supply for respiratory distress patients.

### 4.2. Design of Set-Point Modulated Fuzzy PI-Based MRAC

A hybrid fuzzy PI logic is adopted for automatic gain adjustment of the controller. The two dynamic parameters (error and change of error) are used as the input parameters of the fuzzy system, as shown in Figure 9. 

#### 4.2.1. Fuzzy PI for Adaptive Gain (*m*) Adjustment

It is seen from Figure 9 that the output of the designed fuzzy PI model (*m*) is a function of the input variables (*e*, Δ*e*) and linguistic if-then rules. Here m is used as the alternative of the adaptation gain (*γ*_1_) in the case of MRAC. A very simple, linear, equal base width, and widely used triangular type of fuzzy membership functions with the span of [−100, +100] are used for the inputs (*e* and Δ*e*) and the span of [−1, +1] for output (*m*), and five fuzzy regions (termed negative big (NB), negative medium (NM), zero (ZE), positive medium (PM), and positive big (PB)) are used to develop the database and rule base of the proposed fuzzy scheme as shown in Figure 10 and Figure 11 [30,31,32,33].

The adaptation gain ‘*m*’ as per Equation (24) depends on the input variables (*e*, Δ*e* and the rule base, which consists of 25 fuzzy linguistic if-then rules shown in Table 4. The control parameter ‘θ’ is derived by applying Equation (25) [30].
(24)m=f (e ,Δe) 
(25)$$\theta =\left(-1\right)\times m\times y=-my_{m}$$

The analysis of the control surface of fuzzy PI (*e*, Δ*e* vs. *m*) reveals the smoothness of the surface, which is very much essential for the smooth operations of the control equipment present in the loop. The study of Figure 12 and Table 5 discloses that the output (*m*) of the system depends on the input system parameters (*e*, Δ*e*). The capability of the controller is established from the analysis conducted in Table 5. In this design, the adaptation gain (*γ*_1_) of MRAC is replaced by *m*, which auto-tunes the control parameter (*θ = −my_m_*), for the right amount of oxygen supply to the patients. The effectiveness of the proposed controller is further enhanced by incorporating the set-point modulated scheme with fuzzy PI gain adjustment [28,29]. 

#### 4.2.2. Proposed Scheme (SFPIMRAC)

The final control output u, as shown in Figure 11, is derived as the product of control output (*θ*) and dynamic set-point *ŕ*
(26)$$u=r^{\prime }\times \theta$$

But the dynamic set point
(27)$$r^{\prime }=r-e$$

In the above Equations (26) and (27), the original set-point and error variables are symbolically represented as ‘*r*’ and ‘*e*’, respectively. The consequence of the new dynamic set-point on the proposed system is tabulated in Table 6.

From Equations (25) and (26), it is seen that due to the dynamic variation of the inputs (*e*, Δ*e*) of fuzzy PI, the adaptation gain (*m*) and final control output (*u*) are also altered accordingly [28,29]. As the reference set-point (*r*) is a fixed value, the dynamic set-point (*ŕ*) is modulated only by the temporary increase or decrease in the process error (*e*). This mechanism of set-point variation is used to boost the system’s performance. It is seen from Table 6 that if the error is zero (steady-state condition), there is no variation in the dynamic set-point because *ŕ* = *r*. In cases of undershooting or at the beginning, when the process variable tries to catch the set-point, the error is positive during that period, which indicates that the system needs more oxygen supply, which can be conducted by increasing the set-point ($r^{\prime }=r+e$), which in turn increases the control output ($u=r^{\prime }\times \theta$).With the same principle, in the case of overshooting, the error is negative, and that helps to decrease the control output.

## 5. Results and Discussion

The effectiveness of the proposed controller (SFPIMRAC) is investigated to control oxygen supply to respiratory distress patients. The proposed simulation work is demonstrated using a software tool (MATLAB R2009/SIMULINK) with the following reference (TR_M_) and process model (TP_M_).
$$\begin{array}{c} \mathrm{Reference}\;\mathrm{Model}:\;TR_{M}=\frac{1}{S^{2}+2.5S+1}\\ \mathrm{Process}\;\mathrm{Model}:\;TP_{M}=TF_{M}\times TF_{BO_{2}} \end{array}$$

As far as patient conditions are concerned, the requirement for oxygen concentration varies from patient to patient, from mild to severe. The SpO_2_ reading in the oximeter varies depending on the severity of the infection of the respiratory tract, and the reading in severe respiratory distress patients falls below 90%. In this context, to show the effectiveness of the proposed controller, two set points (90% and 95%) are pondered in this study.

The open-loop step excitation response for both the normal respiratory model and the infected respiratory model is investigated in Figure 13. It is realized that in the case of the developed model, the response never reached the desired level without any control action.

The choice of a proper time constant for a process is a difficult task. A comparative study of the different time constants in terms of the RC value of the oxygen cylinder model is highlighted in Figure 14. The study reveals that at a time constant value of 0.50 s, the developed respiratory model performs better compared to other values. 

Moreover, the performance of MRAC with different adaptive gains (*γ*_1_) is investigated in the developed model. The adaptive gain in MRAC and its consequent effects are observed in Figure 15. 

A change in adaptation gain in MRAC makes the system highly oscillatory; as a result, it fails to reach the desired value quickly, which could be lethal for respiratory distress patients. Sometimes an excess supply of oxygen may create an unhealthy situation. The exact adjustment of adaptation gain in MRAC is a crucial problem, whereas in the SFPIMRAC, there is no need for any human intervention to control the oxygen concentration properly for the respiratory distress patient. The responses for conventional PID, MRAC, and SFPIMRAC are compared. The analysis of Figure 16 (set point 95% SpO_2_) and Figure 16a (set point 90% SpO_2_) reveals that the proposed SFPIMRAC is far more suitable than MRAC and conventional PID in severe respiratory distress patients_._ The variations in oximeter readings (% SpO_2_) did not impact the proposed controller performance divulged in Figure 16 and Figure 17. The model’s performance with variations in the set point (oximeter reading) and time delay are also depicted in Figure 18 and Figure 19. A PID controller is not suggested here, as over-concentration of oxygen can be lethal to patients considered here. The SFPIMRAC response is better than the MRAC response in terms of faster response, low settling time, and no overshoot, which is the most desirable condition in respiratory distress treatment.

The parameters of the human respiratory system model may change depending on the types of obstructive and restrictive lung diseases. The effectiveness of the proposed controller is tested with model parameter variations successfully with minimum deviations in responses. The original gas exchange model (TF_BO2_) is expressed in Equation (15): $${\mathrm{TF}}_{{\mathrm{BO}}_{2}}=\frac{20s^{2}+200s+62}{s^{3}+110s^{2}+350s+67}.$$

The responses of three different models (model1, model2, and model3) are shown with slight parameter variations in Equation (15), as observed in Figure 18 for a reference value of 95%SpO_2_. The convincingness of the proposed controller is further established even with human respiratory model parameter variations in Figure 18.
$${\mathrm{TF}}_{{\mathrm{BO}}_{2}}\left(\mathrm{model}\;1\right)=\left(10s^{2}+300s+62\right)/\left(s^{3}+110s^{2}+350s+67\right)$$
$${\mathrm{TF}}_{{\mathrm{BO}}_{2}}\left(\mathrm{model}\;2\right)=\left(20s^{2}+400s+62\right)/\left(s^{3}+110s^{2}+350s+67\right)$$
$${\mathrm{TF}}_{{\mathrm{BO}}_{2}}\left(\mathrm{model}\;3\right)=\left(20s^{2}+400s+52\right)/\left(s^{3}+110s^{2}+350s+67\right)$$

### 5.1. SFPIMRAC Response with Time Delay at Input and Set-Point/Load Variations

The effects of time delay and set-point/load variations at the input of the process (respiratory model of a severe patient, TP_M_) are observed in Figure 19 and Figure 20, respectively. 

Figure 19 revealed that a time delay in patients’ treatment is not a good idea, though the proposed controller streamlines the flow within a few seconds. During treatment, if the SpO_2_ level falls below the desired value (suppose 95% to 80% for 20 s, as shown in Figure 20) for any unwanted reason, the proposed controller can deliver the required oxygen again immediately without any human intervention, even with the presence of a time delay. 

### 5.2. SFPIMRAC Response with Atime Delay between Two Models and Set-Point/Load Variation

Figure 21 depicts the proposed controller being tested with the concept of time delay (delay due to gas exchange in infected alveolar conditions) in between primary and secondary respiratory models. The typical time delay for human respiratory gas exchange is 0.75 s at rest (transport delay of 0.30 s for inspiration and 0.45 s for expiration), and during exercise, this value comes down to 0.25 s [10]. Given this different time delay in different conditions, the proposed controller is tested with 0.30 s, 0.50 s, and 0.70 s intermediate time delays. The proposed SFPIMRAC can counteract the gas exchange time variations and even load/set-point variation, as successfully shown in Figure 22. 

The proposed controller model can be useful for other simulated and real-time processes. The real-time practical implementation of the developed controller was already demonstrated successfully on a laboratory-based overhead crane [Make: FEEDBACK, UK] for both position and swing control [29].

The control of load pendulation in an overhead crane is a very difficult task. The control strategy for the swing angle control of the overhead crane is implemented and shown in Figure 23 [29]. A comparative study of the set-point modulated fuzzy PD-based model reference adaptive controller (SFMRAC) with the other controllers for the overhead crane model to control swing angle is illustrated in Figure 24 [29].

## 6. Conclusions

An integrated mathematical model of the human respiratory system is presented in this paper for the treatment of a critical patient suffering from respiratory distress. The consideration of inherent time delays caused by the gas exchange between alveolar air space, lung tissue, and capillary blood cells added a new dimension to this study. The efficacy of the mathematical model is verified with model parameter variations, set point changes, inserting different time delays, and stability analysis. The whole study was performed in a MATLAB Simulink environment, and we performed a spirometry test in the laboratory for future studies. The human respiratory system itself exhibits the properties of regulatory control. However, this inherent regulation of the respiratory system sometimes fails due to the attack of various viruses. One of the most popular treatments for patients with lung infections is oxygen support. In this paper, for the proper regulation of external oxygen, a new control strategy is proposed based on the model reference adaptive controller. However, the effective control by MRAC purely depends on the adjustment of the adaptive gain factor. To remove this inherent problem of gain adjustment, a fuzzy knowledge-based system is initiated for the dynamic change of adaptive gain that varies with variations of respiratory system parameters. Two useful concepts are combined to improve controller design: one is a fuzzy-based MRAC scheme, and the other is a dynamic variation of the set-point that is used to automatically track variations in the process parameter. The proposed controller, SFPIMRAC, is tested on different respiratory conditions effectively, and its performance is judged with various time delays and also with load changes. All these advantages of the proposed controller make it suitable for the supplement of oxygen to respiratory distress patients suffering from acute breathing problems. In this context, it is very difficult to conclude about the effectiveness of oxygen therapy and control, but surely it can be used as a supportive treatment. Although respiratory distress of the respiratory system appears to be a complex disease, this research provides some promising avenues for future research.

## Figures and Tables

**Figure 1 diagnostics-13-00967-f001:**
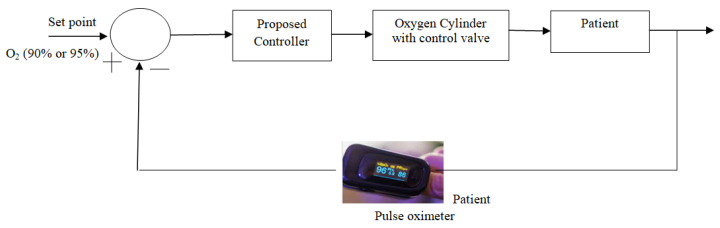
Basic block diagram of the proposed model.

**Figure 2 diagnostics-13-00967-f002:**
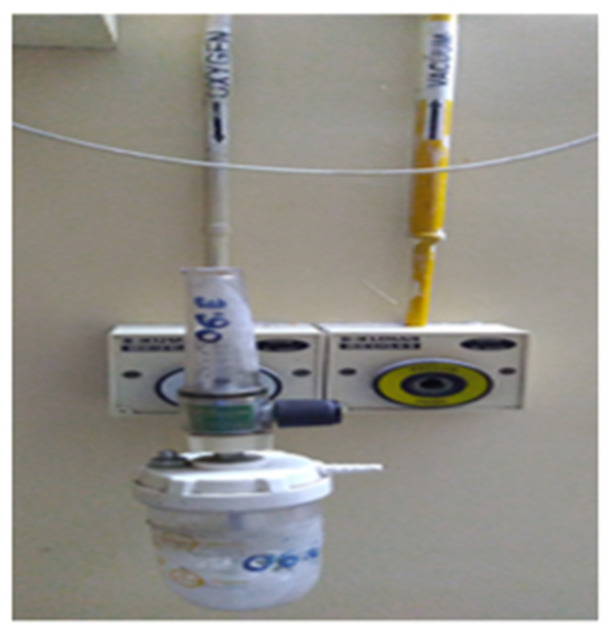
Oxygen supply system near to patient side.

**Figure 3 diagnostics-13-00967-f003:**
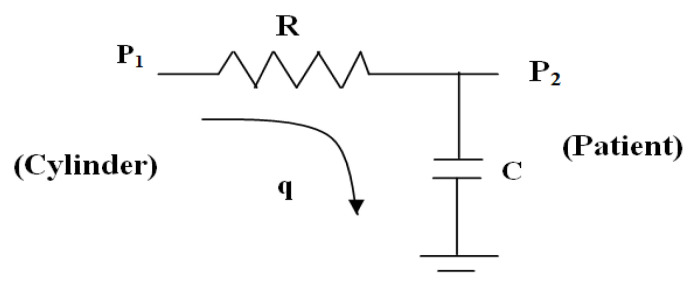
Oxygen cylinder model with input–output.

**Figure 4 diagnostics-13-00967-f004:**
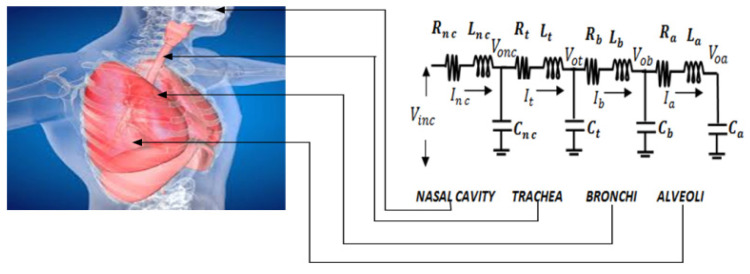
Equivalent electrical R, L, and C model of the human respiratory system.

**Figure 5 diagnostics-13-00967-f005:**
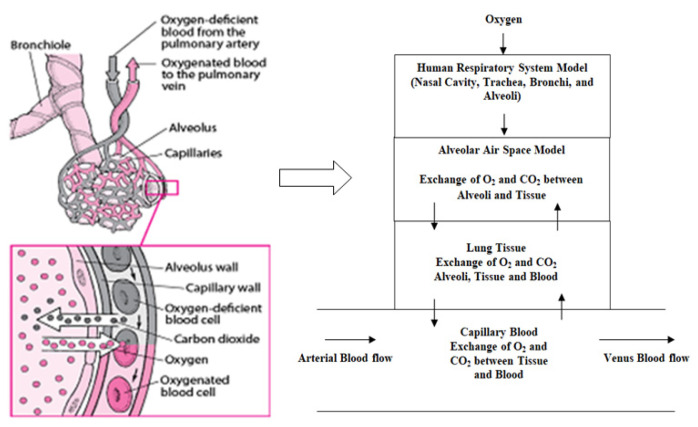
Schematic representation of the simple oxygen exchange model.

**Figure 6 diagnostics-13-00967-f006:**
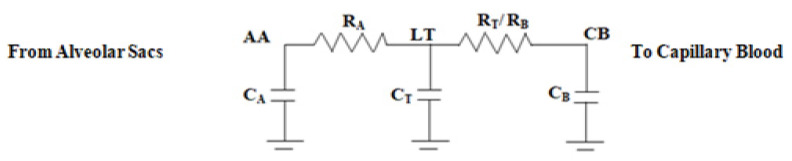
Electrical equivalent circuit diagram of the gas exchange model.

**Figure 7 diagnostics-13-00967-f007:**
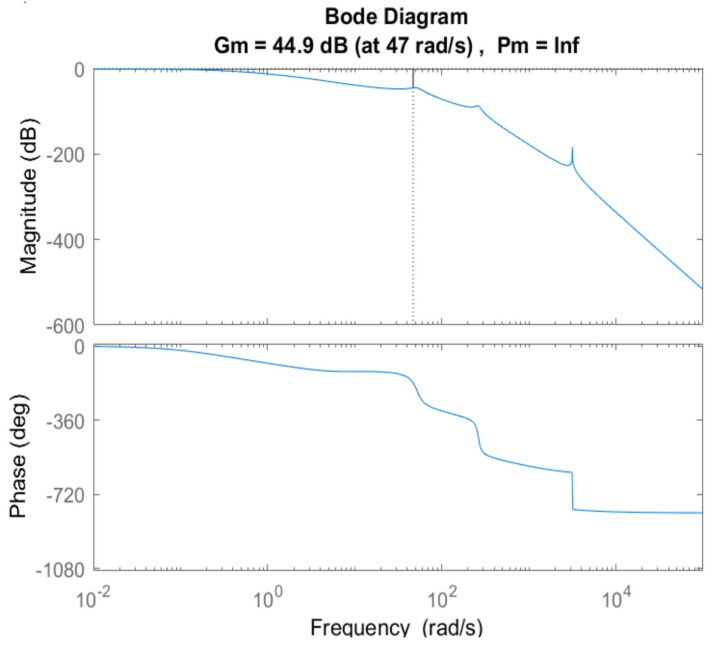
Stability analysis of the proposed model.

**Figure 8 diagnostics-13-00967-f008:**
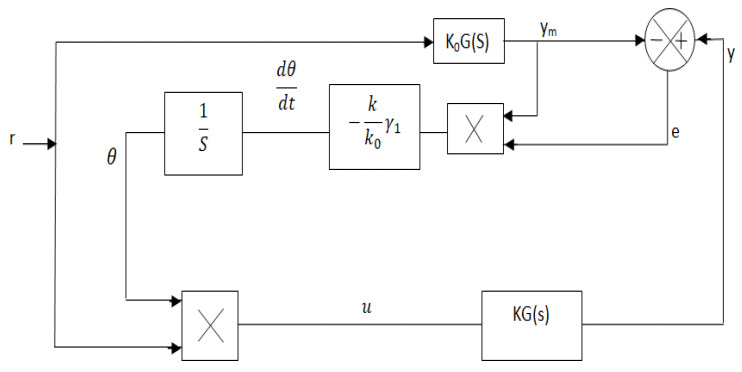
MRAC model to control oxygen concentration for acute respiratory distress patients.

**Figure 9 diagnostics-13-00967-f009:**
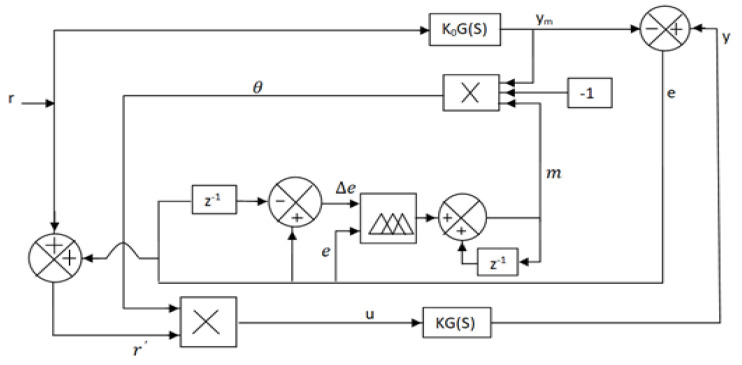
Proposed SFPIMRAC model.

**Figure 10 diagnostics-13-00967-f010:**
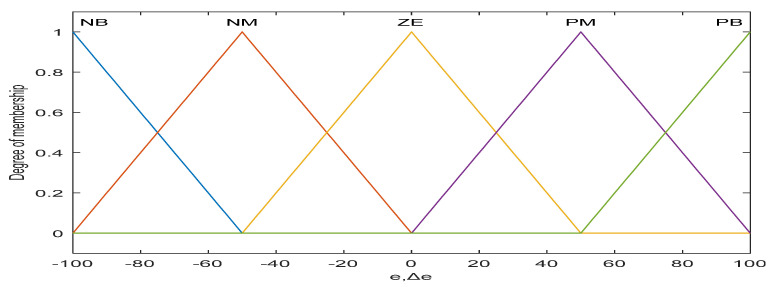
MFs for e and Δe.

**Figure 11 diagnostics-13-00967-f011:**
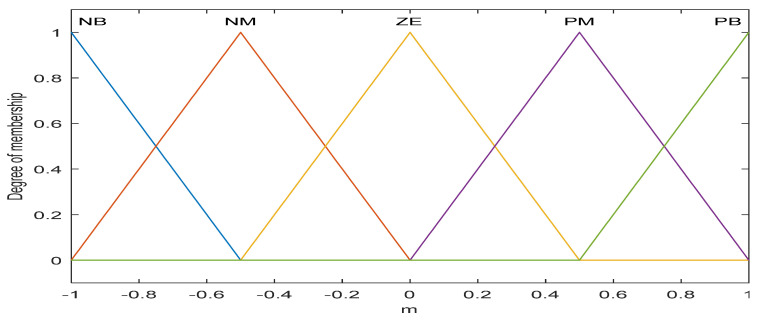
MFs for adaptation gain (m).

**Figure 12 diagnostics-13-00967-f012:**
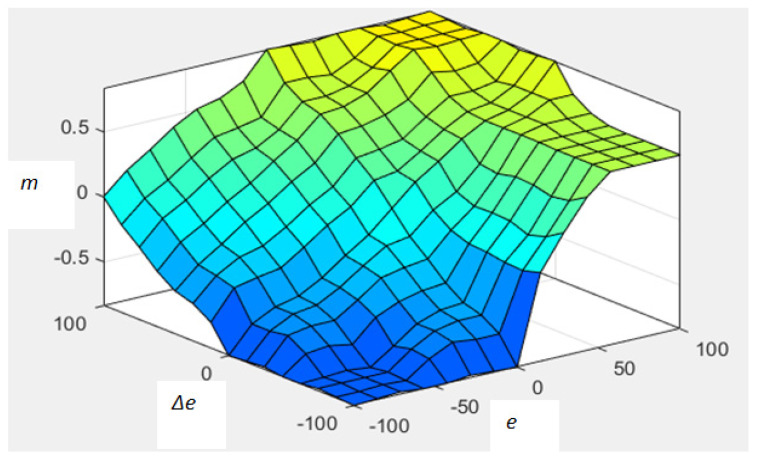
The control surface of fuzzy PI (e, Δe vs.m).

**Figure 13 diagnostics-13-00967-f013:**
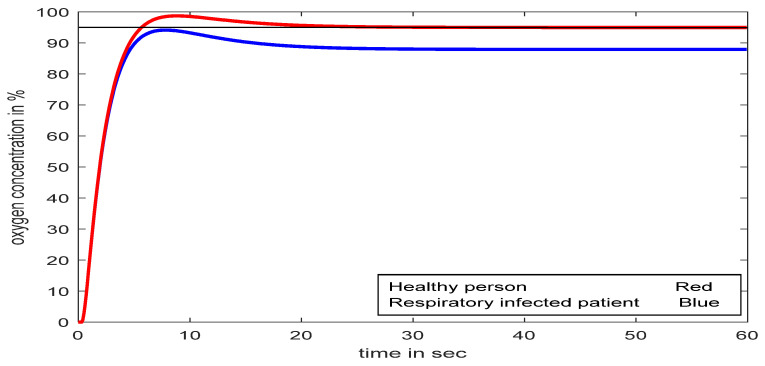
Comparative study of healthy and infected patient with step input at 95%.

**Figure 14 diagnostics-13-00967-f014:**
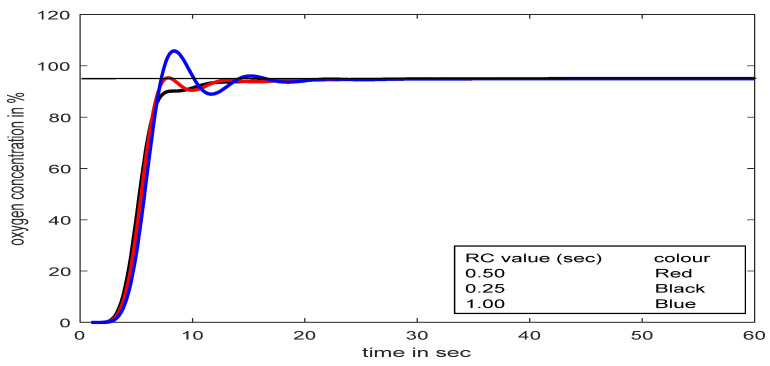
Comparative studies of different time constants of oxygen cylinder model with the proposed controller.

**Figure 15 diagnostics-13-00967-f015:**
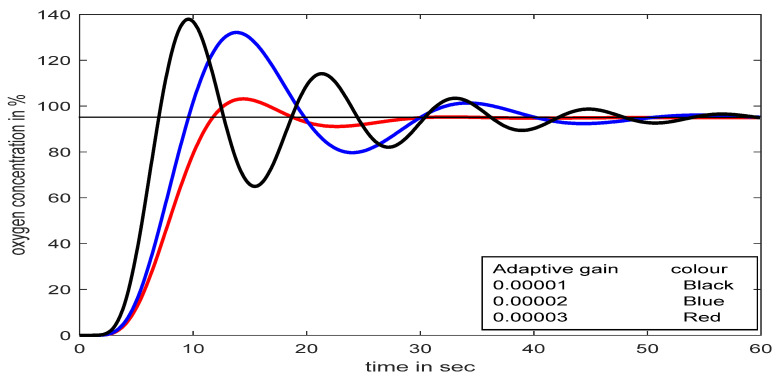
Responses with MRAC for set-point at 95% SpO_2_.

**Figure 16 diagnostics-13-00967-f016:**
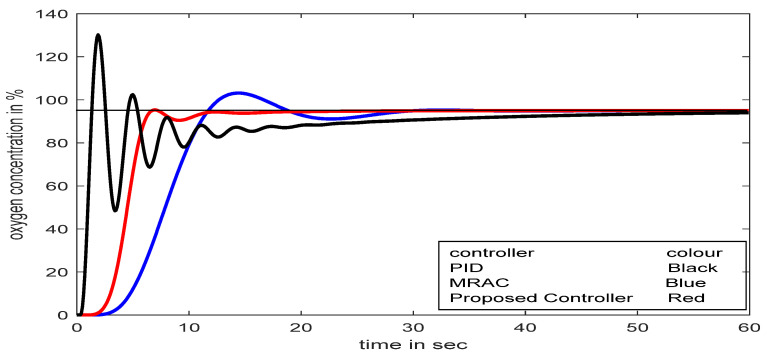
Comparative studies between PID, MRAC, and SFPIMRAC with set-point at 95% SpO_2_.

**Figure 17 diagnostics-13-00967-f017:**
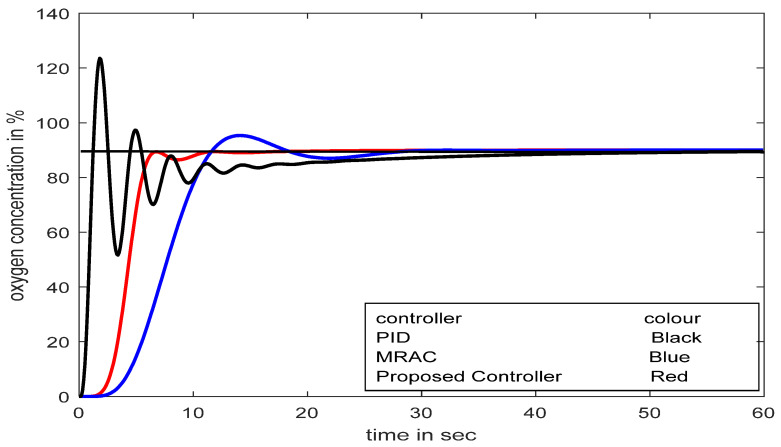
Comparative studies between PID, MRAC, and SFPIMRAC with set-point at 90% SpO_2_.

**Figure 18 diagnostics-13-00967-f018:**
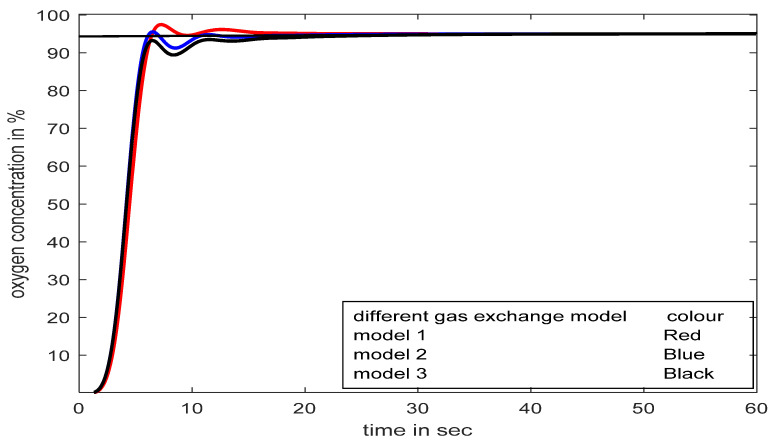
Process responses with SFPIMRAC for model parameters variations.

**Figure 19 diagnostics-13-00967-f019:**
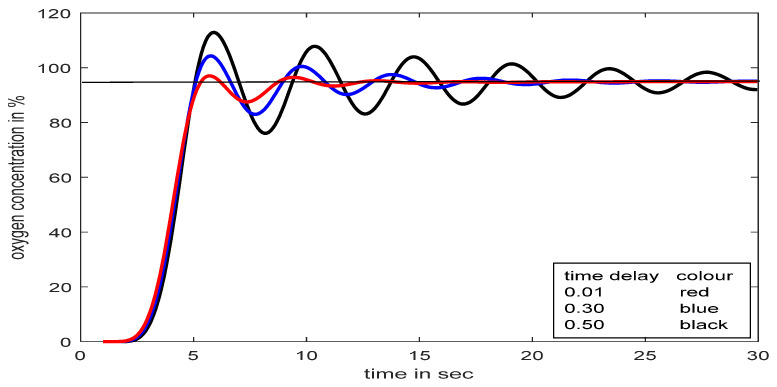
Process responses with SFPIMRAC with different time delay.

**Figure 20 diagnostics-13-00967-f020:**
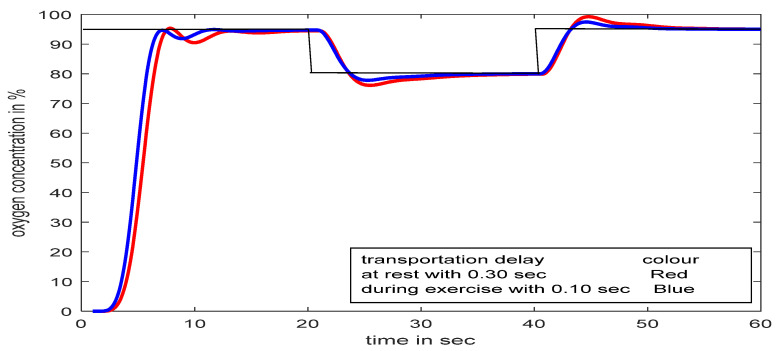
Process responses with SFPIMRAC for time delay and load/set-point variations.

**Figure 21 diagnostics-13-00967-f021:**
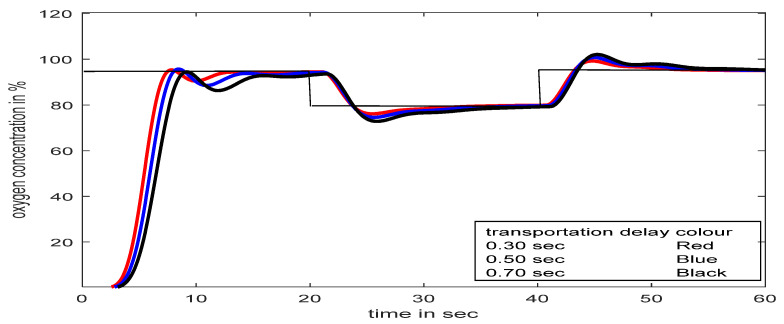
Response with SFPIMRAC for transport delay and load/set-point variations.

**Figure 22 diagnostics-13-00967-f022:**
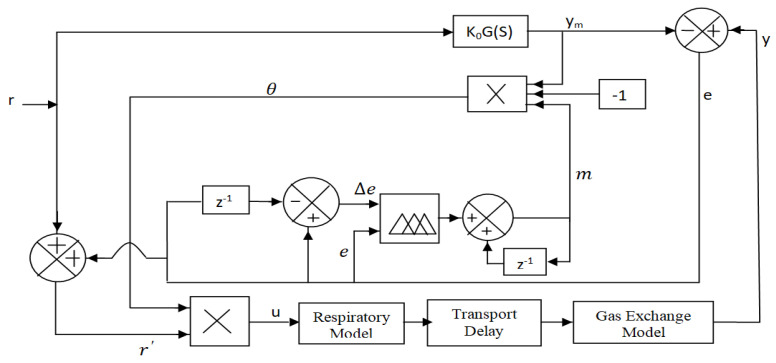
Simulink model of SFPIMRAC with transport delay in between respiratory model and gas exchange model.

**Figure 23 diagnostics-13-00967-f023:**
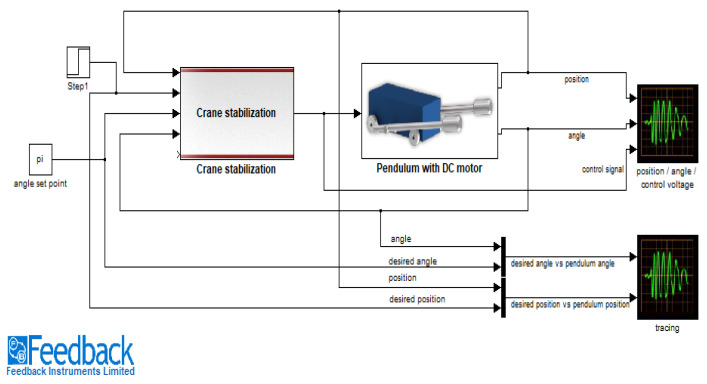
Design of SFMRAC for swing angle control in MATLAB Simulink.

**Figure 24 diagnostics-13-00967-f024:**
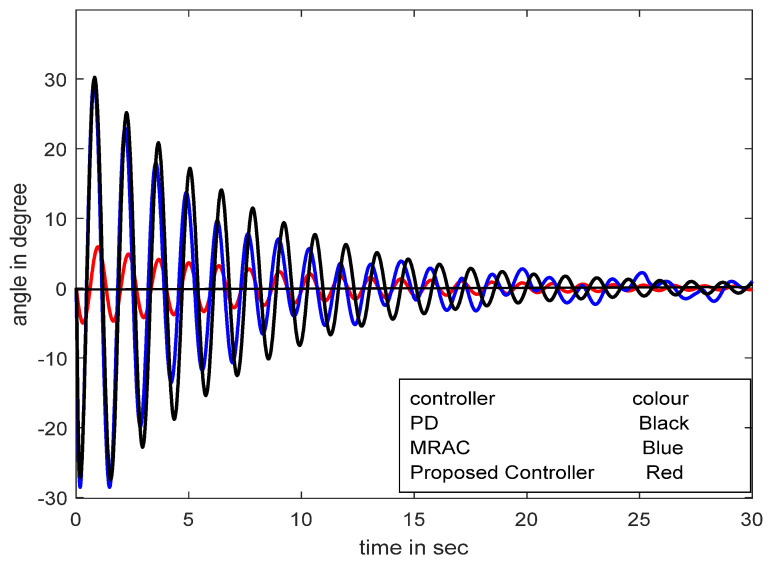
Performance response with PD, MRAC, FPDC, and SFMRAC in swing angle control.

**Table 1 diagnostics-13-00967-t001:** Geometrical dimensions of the morphological model of the respiratory system.

Generation Number	Number of Airways Per Generation	Airways Diameter in cm	Length in cm	Total Airways Area in cm^2^	Avg. Air Flow Velocity in cm/s	Resistance in cm of H_2_O/L ltr./s Calculated Using Formula	Inertance in cm of H_2_O/L/s^2^ Calculated Using Formula	Compliance in ltr./cm of H_2_O Calculated Using Formula
z	n(z)	d = 2r	l	s	u	8 μL/πr^4^	ρL/s	ls/ρn(z)u^2^
0	1	1.8	12	2.54	197	0.0086	0.0059	0.06311
1	2	1.22	4.76	2.33	215	0.008	0.0025	0.0964
2	4	0.83	1.9	2.13	236	0.0075	0.0011	0.0145
3	8	0.56	0.76	2	251	0.0072	0.0004	0.0024
4	16	0.45	1.27	2.48	202	0.0145	0.0006	0.0038
5	32	0.35	1.07	3.11	161	0.0167	0.0004	0.0032
6	64	0.28	0.9	3.96	126	0.0172	0.0002	0.0028
7	128	0.23	0.76	5.1	98	0.0159	0.0001	0.0025
8	256	0.186	0.64	6.95	72	0.0157	0.0001	0.0026
9	512	0.154	0.54	9.56	52	0.0141	7.03 × 10^−5^	0.0029
10	1024	0.13	0.46	13.4	37	0.0118	4.27 × 10^−5^	0.0035
11	2048	0.109	0.39	19.6	26	0.0101	2.48 × 10^−5^	0.0044
12	4096	0.095	0.33	28.8	17	0.0074	1.43 × 10^−5^	0.0064
13	8192	0.082	0.27	44.5	11	0.0054	7.55 × 10^−6^	0.0097
14	16,384	0.074	0.16	69.4	7.2	0.0024	2.87 × 10^−6^	0.0105
15	32,768	0.05	0.13	117	4.3	0.0048	1.41 × 10^−6^	0.0206
16	65,536	0.049	0.11	225	2.2	0.0022	6.19 × 10^−7^	0.0638
17	131,072	0.04	0.09	300	1.7	0.002	3.86 × 10^−7^	0.0591
18	262,144	0.038	0.08	543	0.92	0.0011	1.90 × 10^−7^	0.1632
19	524,288	0.036	0.07	978	0.51	0.0005	8.91 × 10^−8^	0.4034
20	1,048,576	0.034	0.07	1740	0.29	0.0003	5.01 × 10^−8^	1.1099
21	2,097,152	0.031	0.07	2730	0.18	0.0002	3.19 × 10^−8^	2.26
22	4,194,304	0.029	0.67	5070	0.99	0.0016	1.64 × 10^−7^	0.664
23	8,388,608	0.025	0.07	7530	0.66	0.0001	1.24 × 10^−8^	0.1241

**Table 2 diagnostics-13-00967-t002:** R, L, and C values.

Different Section of Human Respiratory System	R H_2_O/ltr/s	L H_2_O/ltr/s^2^	C ltr/cm of H_2_O	RC	LC
NASAL CAVITY	16.332700	0.0200000000	0.1320	2.156000	0.00270000000
TRACHEA	0.086000	0.0059000000	0.0631	0.005400	0.00037000000
BRONCHI	0.008700	0.0002929000	0.0461	0.000402	0.00001440000
ALVEOLI	0.000550	0.0000000647	1.0396	0.000571	0.00000006720

**Table 3 diagnostics-13-00967-t003:** Parameters values of the gas exchange model.

Parameters	Alveolar Air	Tissue	Capillary Blood
Volume (L)	V_A_ = 1.9 × 10^−7^	V_T_ = 4.2 × 10^−8^	V_B_ = 7.5 × 10^−9^
Molar concentration (M/mm)	σ_A_ = 2.5 × 10^−5^	σ_A_ = 1.2 × 10^−6^	σ_B_ = 1.2 × 10^−6^
Diffusion rate (L/s)	D_TA_ = 2.4 × 10^−12^	D_TA_ = (6.7 – 10) × 10^−12^	

**Table 4 diagnostics-13-00967-t004:** Rules for computing fuzzy output.

e/Δ*e*	NB	NM	ZE	PM	PB
NB	NB	NB	NB	NM	ZE
NM	NB	NB	NM	ZE	PM
ZE	NB	NM	ZE	PM	PB
PM	PM	PM	PM	PB	PB
PB	PM	PM	PB	PB	PB

**Table 5 diagnostics-13-00967-t005:** Study of control surface (Figure 12).

e = (y − y_m_)	Δ*e*	m	Output (θ = −my_m_)
+ve (high); y > y_m_	+ve (high)	+ve (high)	−ve/ need to decrease oxygen flow as y > y_m_
0; y = y_m_	0	0	No change in oxygen flow, maintain the same flow
−ve (high); y < y_m_	−ve (high)	−ve (high)	+ve/ need to increase oxygen flow as y < y_m_

**Table 6 diagnostics-13-00967-t006:** Mathematical representation of set-point variations.

e	r’ = r + e	Remarks
+ve	r’→+ve	r’ > r
−ve	r’→+ve	e < r and r’ < r
zero	r’ = r	r’ = r

## Data Availability

Not applicable.

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
