# Peer review of "Self-Regulating Adaptive Controller for Oxygen Support to Severe Respiratory Distress Patients and Human Respiratory System Modeling"

_diagnostics, 2023, doi:10.3390/diagnostics13050967_

Round 1

Reviewer 1 Report

The article titled “Self-Regulating Adaptive Controller for Oxygen Support to Severe Respiratory Distress Patients and Human Respiratory System Modeling” proposes a methodology of Fuzzy based emergency oxygen support system focusing towards the respiratory diseases. The method covers the detailed design technique of the Fuzzy PI controller with comprehensive mathematical modeling of the respiratory system. However, the following concerns may be addressed for better understanding of the presented work:

1. Though the requirement of the proposed work is explained, the article lacks to address the state of the art of the area under research.

2. The main contribution(s) of the proposed work based on the limitations of the earlier works may be added in a separate section under section 1 (Introduction).

3. The study does not consider the obstructive and restrictive lung disease models for which the work is focused. Or, it is required to present the capacity of the proposed model to consider the above mentioned disease conditions.

4.  The statement mentioned in line number 165 and 166 should be supported by a reference or the basis of the demand in the statement requires to be addressed.

5. An outline for practical implementation of the proposed method may be added to enhance the significance of the work.

6. The speed of response of the overall system may be tabulated with simulated variation in oximeter reading can be introduced for the clarity of the result.

Author Response

To

Diagnostics Editorial Office

Ref:  email dtd. 02/01/2023 (invitation paper) from Prof. Dr. Fleur T. Tehrani 

Ref.: Manuscript ID: diagnostics-2170721

Dear editors’ and referees,

Sub. : Minor Revision of Manuscript ID: diagnostics-2170721

Hope you are doing great. With reference to the above subject and the observations made by the reviewers, we brought some changes in the manuscript as follows:

Reviewer

Comments

Corrections/Modifications

Reviewer 1

1. Though the requirement of the proposed work is explained, the article lacks to address the state of the art of the area under research.

The area under research is addressed in Abstract portion by explaining the novelty of the research. (page 1)

2. The main contribution(s) of the proposed work based on the limitations of the earlier works may be added in a separate section under section 1 (Introduction).

Same is addressed with reference in introduction part. (page2)

3. The study does not consider the obstructive and restrictive lung disease models for which the work is focused. Or, it is required to present the capacity of the proposed model to consider the above mentioned disease conditions.

Model of the respiratory system would be different for different type of respiratory diseases. We studied the effectiveness of the controller in respiratory model with parameter variations. (page 16 & 17, Figure 16b)

4. The statement mentioned in line number 165 and 166 should be supported by a reference or the basis of the demand in the statement requires to be addressed.

We added some references.

5. An outline for practical implementation of the proposed method may be added to enhance the significance of the work.

Added the details of practical

implementation in page 18 with journal  

 reference.

6. The speed of response of the overall system may be tabulated with simulated variation in oximeter reading can be introduced for the clarity of the result.

System response is checked with variations of Oximeter reading shown in Figure 16 a. (page 15 & 16)

We hope you all find the above changes effective and suitable for your renowned journal. Please inform us of any further clarification.

With best regards,

Arabinda Kumar Pal & Indrajit Naskar

Reviewer 2 Report

I am really grateful to review this manuscript. In my opinion, this manuscript can be published once some revision is done successfully. This study developed a self-regulating adaptive controller for oxygen support to severe respiratory distress patients. This study achieved this aim based on rigorous mathematical modeling and simulation. I would argue that this is a rare achievement. However, it needs to be noted that every mathematical modeling and simulation has an issue of external validation. I would like to suggest the authors to address this important issue in the section of Conclusions. 

Author Response

To

Diagnostics Editorial Office

Ref:  email dtd. 02/01/2023 (invitation paper) from Prof. Dr. Fleur T. Tehrani 

Ref.: Manuscript ID: diagnostics-2170721

Dear editors’ and referees,

Sub. : Minor Revision of Manuscript ID: diagnostics-2170721

Hope you are doing great. With reference to the above subject and the observations made by the reviewers, we brought some changes in the manuscript as follows:

Reviewer

Comments

Corrections/Modifications

Reviewer 2

It needs to be noted that every mathematical modeling and simulation has an issue of external validation. I would like to suggest the authors to address this important issue in the section of Conclusions

We addressed this issue in conclusion as advised (page 19)

We hope you all find the above changes effective and suitable for your renowned journal. Please inform us of any further clarification.

With best regards,

Arabinda Kumar Pal & Indrajit Naskar